# The True Nature of Tricalcium Phosphate Used as Food Additive (E341(iii))

**DOI:** 10.3390/nano13121823

**Published:** 2023-06-08

**Authors:** Youssef El Moussaoui, Hélène Terrisse, Sophie Quillard, Marie-Hélène Ropers, Bernard Humbert

**Affiliations:** 1Nantes Université, CNRS, Institut des Matériaux de Nantes Jean Rouxel, IMN, F-44000 Nantes, France; helene.terrisse@cnrs-imn.fr (H.T.); sophie.quillard@cnrs-imn.fr (S.Q.); 2INRAE, UR1268 Biopolymères Interactions Assemblages, F-44316 Nantes, France; marie-helene.ropers@inrae.fr

**Keywords:** hydroxyapatite, E341, characterization, tricalcium phosphate food additive, nanoparticles

## Abstract

Tricalcium phosphate (TCP) is a food additive, labeled E341(iii), used in powdered food preparation, such as baby formula. In the United States, calcium phosphate nano-objects were identified in baby formula extractions. Our goal is to determine whether the TCP food additive, as is used in Europe, can be classified as a nanomaterial. The physicochemical properties of TCP were characterized. Three different samples (from a chemical company and two manufacturers) were thoroughly characterized according to the recommendations of the European Food Safety Authority. A commercial TCP food additive was identified as actually being hydroxyapatite (HA). It presents itself in the form of particles of different shapes (either needle-like, rod, or pseudo-spherical), which were demonstrated in this paper to be of a nanometric dimension: E341(iii) is thus a nanomaterial. In water, HA particles sediment rapidly as agglomerates or aggregates over a pH of 6 and are progressively dissolved in acidic media (pH < 5) until the complete dissolution at a pH of 2. Consequently, since TCP may be considered as a nanomaterial on the European market, it raises the question of its potential persistency in the gastrointestinal tract.

## 1. Introduction

Nanomaterials, which are defined as materials having at least one dimension below 100 nm [1], have become ubiquitous in our daily lives. Their presence as nanoparticles is well-known in common consumer products, such as cosmetics, various pharmaceutical products, textiles and food products [2,3,4,5]. Nanomaterials remain a source of concern because they can exhibit very different properties from those of their bulk form. The high specific surface area of nanomaterials often heightens their reactivity and their small dimensions can lead to cell internalization in the event of their absorption through the gastrointestinal tract [6]. Over the last decade, several European national registers have begun data collection on nanomaterials used in food products [7]. In particular, food additives should be examined, as they are likely to occur in multiple types of food products. In the United States, calcium phosphate nanoparticles have been identified in baby formula, but their source has not been attributed to any specific food additive [8]. In Europe, calcium phosphates are used in cosmetics and in food additives (coded as E341) for their anticaking properties as well as for their dietary source of calcium and phosphorus in numerous products, including baby formula [9,10].

The E341 code refers to three different calcium salts of phosphoric acid (H_3_PO_4_): E341(i) (monocalcium phosphate/calcium phosphate monobasic, Ca(H_2_PO_4_)_2_), E341(ii) (dicalcium phosphate/calcium phosphate dibasic, CaHPO_4_), and E341(iii) (tricalcium phosphate (TCP)/calcium phosphate tribasic, Ca_3_(PO_4_)_2_). The E341(iii) notation will be preferred to the TCP one in this paper. The lack of solubility in water of certain calcium phosphate compounds leads to questions concerning their persistency in acidic gastric conditions [11]. As the E341(iii) food additive is also listed in Europe, and is present in baby formula powdered preparations, the investigation of its nanometric nature is a priority.

On the basis of the previous work in the United States [8], the goal of this study is to identify if the E341(iii) food additive, as is commercialized in Europe, fulfills the criteria for a nanomaterial. Our work will aim at characterizing both its crystallographic structure and its physicochemical properties. A number of parameters can influence its behavior in a complex medium, such as the core and surface composition of the particles, their surface electric charge, their aggregation state, morphology, and size [12]. Agencies such as the European Food and Safety Authority (EFSA) provide guidelines to identify materials exhibiting nanoscale dimensions and their physicochemical properties [13]. These recommendations are taken into account to conduct this study.

## 2. Materials and Methods

### 2.1. Chemicals

Three samples of food-grade E341(iii) (labeled A, B, and C) were obtained from different sources and were characterized without any modification of their pristine state. The E341(iii) samples all consist in appearance as a finely divided white powder. E341(iii) A is a commercially available sample, obtained from Sigma Aldrich, which fulfills the E341(iii) definition according to the European Pharmacopeia. Two different industrial manufacturers whose products are suitable for European consumption provided E341(iii) B and E341(iii) C. The water used in all experiments was deionized water (Millipore, Burlington, MA, USA) with a resistivity of 18.2 MΩ·cm. Sodium chloride (>99% purity) was purchased from Acros Organics. Nitric acid (65%) and hydrochloric acid (1 mol.L^−1^) were purchased from Carlo Erba Reagents. Sodium hydroxide (pharma-grade pellets) was purchased from Panreac. Single-element standard solutions (1000 µg/mL) of calcium and phosphorus were purchased from ChemLab.

### 2.2. X-ray Diffraction (XRD)

X-ray diffraction (XRD) was performed using a D8 Bruker diffractometer in the Bragg–Brentano geometry, equipped with a front germanium (111) monochromator, a copper anode (Cu Kα1 radiation; λ = 1.540598 Å), and a LynxEye PSD detector. The scanning consisted of a 2θ angular range from 8° to 90° and a step size of 0.018°. The obtained diffraction pattern was analyzed using the Diffrac.eva V5.2 software for phase assignment according to the Powder Diffraction File (PDF) database. A Rietveld refinement was made using the Jana2006 software to determine the proportion of phases in the E341(iii) A sample.

### 2.3. Raman Spectroscopy

Raman spectra were obtained using a Renishaw InVia Reflex spectrometer coupled with a Leica optical microscope. The instrument was equipped with a double-edge filter to eliminate the Rayleigh scattering and with a CCD camera working at a temperature of 220 K with a 1024 × 246 pixel array. The spectral resolution achieved with the use of gratings of 2400 grooves/mm was 3 cm^−1^. The wavenumber accuracy in a vacuum was better than 0.8 cm^−1^. All spectra were recorded in the 300–1100 cm^−1^ range with an argon ion laser source emitting a 514.53 nm excitation wavelength. Inelastic spectra were collected in the backscattering mode via the confocal optical microscope. A ×50 magnification objective allowed a precise area analysis with a focused laser power of 1 mW/µm^2^. The probed area was around 4 µm^2^. The powder sample was deposited onto a glass pure silica slide. The powder surface was flattened with a flat spatula to have a well-defined focal distance. Measurements were performed on several points of the sample to ensure homogeneity and the focused power of the laser beam was checked for each wavelength to avoid any transformation or heating of the sample.

### 2.4. Solid State ^31^P Nuclear Magnetic Resonance (NMR)

The ^31^P MAS (magic angle spinning) NMR spectra were acquired on a Bruker NEO 300 MHz (7 T) spectrometer. The acquisition was made under ^1^H decoupling and consisted of a single *π*/2 excitation pulse with a length of 2.5 µs. A recycling delay of 60 s ensured quantitative spectra. A 4 mm MAS probe was used. The powders were introduced in a ZrO_2_ rotor and submitted to a 10 kHz spin speed at MAS. H_3_PO_4_ was used as reference.

### 2.5. Infrared Spectroscopy

Infrared spectra were recorded on a Fourier transform spectrometer (FT-IR) Bruker Vertex 70 equipped with a Specac attenuated total reflectance (ATR) accessory. The ATR mode enabled the study of pure E341(iii), contrary to the transmission mode, which often requires the preparation of a mixture of the sample powder with a nonabsorbing IR material (e.g., KBr) to yield pellets, which were then analyzed. The diamond crystal enabled a single reflection measurement. Acquisitions were performed over 100 scans in the 400–6000 cm^−1^ range with a 4 cm^−1^ resolution. The presentation of the spectra was obtained by using the Opus software V7.5 and did not undergo any modification. The reflection mode, compared to the transmission mode, induced shifts of the maximum wavenumber and changes of the band profiles [14]. In this paper, all reported wavenumbers for the IR will correspond to the ATR experimental spectra.

### 2.6. Specific Surface Determination

Adsorption and desorption isotherms were performed on a Micrometrics 3Flex adsorption analyzer. The specific surface area and C_BET_ constants of the E341(iii) samples were determined from the Brunauer–Emmett–Teller (BET) analysis of the N_2_ adsorption and desorption isotherm curves at 77 K. The C_BET_ constant (Equation (1)) is related to the heat of adsorption of N_2_ to the surface.
(1)$$C_{BET}=\exp\left(\frac{E_{1}-E_{L}}{RT}\right)$$
where E_1_ is the heat of the adsorption for the first layer, E_L_ is the heat of the adsorption (equivalent to the heat of liquefaction) of the subsequent layers, R is the gas constant, and T is the temperature.

### 2.7. Pycnometry

Pycnometry was performed on a Micrometrics AccuPyc II gas pycnometer. Density was determined at room temperature using helium gas for the volume determination of a given sample mass. 

### 2.8. Transmission Electron Microscopy (TEM)

Transmission electron microscopy (TEM) was performed on a S/TEM Themis Z G3 (Thermo Fischer Scientific) operating at a 300 kV accelerating voltage. The samples were prepared from a solution of E341(iii) in absolute ethanol (50 µL/mL). One drop was then deposited on a lacey carbon copper grid. After solvent evaporation, the grid was used for analysis. Several images taken at different spots of the samples allowed the assessment of the size and shape.

### 2.9. Inductively Coupled Plasma Atomic Emission Spectroscopy (ICP-AES)

Inductively coupled plasma atomic emission spectroscopy (ICP-AES) was performed on a Thermo Scientific iCAP 6300 (LPG UMR 6112) for calcium and phosphorus determination. A cobalt ion standard line was used to ensure stability throughout the sample measurements. The samples were measured in triplicate. Each sample of E341(iii) (20 mg) was dissolved in 1 mL of concentrated nitric acid. Water was added to yield a 100 mL aqueous solution of dissolved E341(iii). The solution was diluted (1:10) before the ICP-AES analysis. Calcium and phosphorus standard solutions (1000 µg/mL) were used to prepare a calibration curve using element emission measurements at 317.7 nm (Ca) and 185.9 nm (P).

### 2.10. Dissolution Determination

E341(iii) (100 mg) was placed under stirring (40 rpm) in 20 mL of ultrapure water for 48 h at room temperature. The pH was set after the addition of E341(iii) and adjusted with NaOH (10^−2^ mol.L^−1^) and HCl (10^−1^ mol.L^−1^) solutions. After 48 h, the remaining solid was recovered by centrifugation (5970× *g*, 20 min) and placed to dry under vacuum in a desiccator. Once dried, the remaining solid was weighed.

### 2.11. Granulometry

The size of the agglomerates formed by the E341(iii) samples in neutral pH conditions at room temperature was determined using a laser diffraction particle size analyzer (Partica LA-960, Horiba, Kyoto, Japan). The instrument allowed the determination of the particle size in the 0.01–5000 µm range using two light sources (650 nm and 405 nm). The measurements were performed using the internal cuvette (15 mL) by filling it with water, which served as a reference before the introduction of an appropriate volume (1–5 mL) of the E341(iii) aqueous solution (0.5 g.L^−1^). An acceptable range of the obscuration percentage (3–5%) for the two light sources needs to be reached to perform the measurements. An alignment was performed before each measurement. The refractive index of hydroxyapatite (1.630) was used and the imaginary part was set at zero.

### 2.12. Zeta-Potential Measurements

The zeta potential of the E341(iii) samples was measured with a Zetasizer Nano ZS (Malvern Instruments Ltd., Malvern, UK) equipped with a He-Ne laser. The scattered light was measured at an angle of 173°. All measurements were made at room temperature after 30 s of equilibration time with 5 runs (acceptable if there were at least 3 satisfactory runs). The results were obtained using the Henry equation after the application of the Smoluchowski approximation. Sodium chloride (0.01 mol.L^−1^) was used as a supporting electrolyte. Suspensions were prepared by introducing E341(iii) in 20 mL of water (0.5 g.L^−1^), and the pH was adjusted to the set value by adding HCl (10^−1^ mol.L^−1^) and NaOH (10^−2^ mol.L^−1^) solutions. Multiple solutions, each one corresponding to a given pH value, were prepared to achieve the different zeta-potential measurements.

## 3. Results

### 3.1. Characterization of the Pristine E341(iii) Food Additive

#### 3.1.1. X-ray Diffraction

XRD patterns (Figure 1) provide information on the composition of the studied E341(iii) powders. E341(iii) A is mostly composed of hydroxyapatite (HA) and contains a fraction of dicalcium phosphate anhydrous (DCPA, also called monetite). A Rietveld refinement found the HA/DCPA proportion to be 82:18. In the case of E341(iii) B and C, it appears that they are only composed of HA.

The crystallite size can be obtained using the Scherrer equation (Equation (2)):(2)$$D=\frac{K\lambda }{\beta cos\Theta }$$
where D is the crystallite size (nm), K is the Scherrer constant, which is shape-dependent (here, K = 0.9), λ is the wavelength of the X-ray beam (here, λ = 0.15406 nm), β is the full width at half-maximum (in radians), and θ is the Bragg angle (in radians).

A crystallite can be defined as a material domain of a given crystalline structure and should not be confused with the particle itself, which may contain several crystallites. The crystallite size was determined for the (002) peak (2θ = 26°) because of its relatively high intensity and the absence of any significant overlapping peak. The crystallite size of E341(iii) A, B, and C was found to be 25 nm, 80 nm, and 40 nm, respectively. 

By adding a known amount of magnesium oxide (MgO), we verified that the E341(iii) A sample, constituted of HA and DCPA, does not contain any amorphous phase [15]. Indeed, after a Rietveld analysis of the obtained XRD pattern, the amount of MgO could be matched to the one introduced, implying there was no amorphous phase in the sample (Appendix A).

The HA and DCPA structures are well-known (Figure 2). HA presents a hexagonal structure containing two types of calcium atoms forming a network with phosphate groups encompassing hydroxyl groups in channels along the c-axis [16]. Additionally, HA possesses only one type of phosphorus atom, while DCPA has two different types of phosphorus. The two phases can therefore be differentiated using ^31^P NMR spectroscopy.

#### 3.1.2. Solid State ^31^P Nuclear Magnetic Resonance (NMR) Spectroscopy

The solid state ^31^P NMR spectroscopy (Figure 3) shows that all three samples contain HA, with its characteristic peak found here at 2.85 ppm. In the case of E341(iii) A, the presence of DCPA was evidenced by the two peaks of the nonequivalent phosphorus sites in the DCPA structure at −1.47 and −0.14 ppm [17].

#### 3.1.3. Infrared and Raman Spectroscopy

The different vibration modes exhibited by HA stem from the symmetry (space group P6_3_/m) of its crystallographic structure. The irreducible representation of HA shows active modes in the IR and Raman spectroscopy (Equation (3)). Vibrational spectroscopy is an additional characterization tool which is more sensitive to secondary phases or trace chemicals than XRD. NMR experiments, which are element-specific, can also be completed by spectroscopy analysis.
Γ_N_ = 13A_g_ + 9B_g_ + 9E_1g_ + 13E_2g_ + 9A_u_ + 13B_u_ + 13E_1u_ + 9E_2u_(3)

The IR measurements were performed using the ATR mode, which does not require the preparation of a KBr pellet, as it is usually the case for the transmission mode (Figure 4). The four phosphate vibration modes, ν_1_ (962 cm^−1^), ν_2_ (470 cm^−1^), ν_3_ (1020 cm^−1^, 1062 cm^−1^, 1090 cm^−1^), and ν_4_ (561 cm^−1^, 571 cm^−1^, 601 cm^−1^), as well as the hydroxyl libration (631 cm^−1^) and stretching modes (3571 cm^−1^), were found to be characteristic of HA in the ATR-FTIR [18].

It is expected to be able to observe corresponding bands on the IR spectra for E341(iii) A containing DCPA. The overlap with HA vibration bands, as well as the lower proportion of DCPA, make these bands difficult to distinguish. However, two bands corresponding to the DCPA phosphate vibration modes ν_4_‴ and ν_3_′ are displayed (shoulders at approximately 530 cm^−1^ and 1120 cm^−1^) [19]. E341(iii) B presents broader band features, most noticeable for the 1120 cm^−1^ band. This could be interpreted as a poorer crystallinity of HA than its E341(iii) A and C counterparts, as the HA ν_1_–ν_3_ region is a good indication of HA crystallinity [20]. 

Besides the infrared active modes (9A_u_ + 13E_2g_), the Raman active modes (13A_g_ + 9E_1g_ + 13E_2g_) also enable the characterization of HA. Indeed, Raman spectroscopy (Figure 5) confirms that E341(iii) A is composed of HA and DCPA, whereas E341(iii) B and C only contain HA. The different HA phosphate vibration modes—ν_2_ (429 cm^−1^, 447 cm^−1^), ν_4_ (579 cm^−1^, 591 cm^−1^, 607 cm^−1^), ν_1_ (962 cm^−1^), and ν_3_ (1028 cm^−1^, 1046 cm^−1^, 1076 cm^−1^)—were identified [21]. Moreover, by using micro-Raman, the determination of specific structures within E341(iii) was made possible. The optical microscope of the Raman spectrometer showed that E341(iii) A contained micrometric plates, which were analyzed apart from the dominating finely divided powder (Figure 5a). The spectra of these micrometric plates, which exhibit bands at 986 cm^−1^ (ν_1 DCPA_) and 901 cm^−1^ (ν_3_‴ _DCPA_), can be associated with the proportion of DCPA that is present in E341(iii) A [22]. Due to the laser size spot (around 4 µm^2^) and the finer powder of HA, the spectra of the DCPA phase contained in E341(iii) A also presents bands corresponding to HA. 

#### 3.1.4. Ca/P Molar Ratio Determined by ICP-AES

Calcium phosphate compounds can be differentiated by their Ca/P molar ratio [23]. ICP-AES (Table 1) allows the determination of the content of Ca and P in the E341(iii) samples. E341(iii) A has a lower value of the Ca/P ratio than E341(iii) B and C due to its content of DCPA. Indeed, the Ca/P ratio of DCPA (1.00) is lower than that of HA (1.67). The HA/DCPA ratio obtained through this method (78:22) is similar to the previous one obtained using the Rietveld method (82:18).

### 3.2. Physicochemical Characteristics (Density and Specific Surface Area) of E341(iii) Powders

The specific surface areas of E341(iii) A and E341(iii) C are similar, with respective values of 58 m^2^/g and 60 m^2^/g (Table 2). E341(iii) B has a much lower specific surface area, with a value of 9 m^2^/g. The C_BET_ constant must also be considered, as it reveals the surface affinity for N_2_ and the overall surface reactivity. E341(iii) A stands out as having a C_BET_ constant twice as high as E341(iii) B. E341(iii) C has a C_BET_ constant value in-between E341(iii) A and B.

The complete N_2_ adsorption/desorption isotherms of the different E341(iii) powders were performed (Figure 6). The samples all displayed a type IV adsorption/desorption isotherm characteristic of mesoporous materials. The most notable difference relies in a lower maximal volume of adsorbed N_2_ (25.4 cm^3^/g) for E341(iii) B, in agreement with its smaller specific surface area. In contrast, E341(iii) A and C show a very similar isotherm, with a maximal adsorbed value of 173.2 cm^3^/g and 172.2 cm^3^/g, respectively.

Another commonly accepted criterion to estimate if a sample is a nanomaterial is the volume specific surface area (VSSA), which is the product of the specific surface area and the density. Above a threshold value of 60 m^2^/cm^3^, the considered material can be defined as a nanomaterial [24]. E341(iii) A and C fulfill this criterion, while for E341(iii) B, its much lower value is not sufficient to rule it out as not being nanometric. Other parameters, such as particle size, have to be considered.

Additionally, the average particle diameter (d_BET,App_) can be estimated using the specific surface area value (Sw) and the density of the powders (ρ). This calculation assumes that the particles are spherical and that all of them present the same size.
(4)$$d_{BET,App[nm]}=\frac{6000}{Sw\left\lfloor \frac{m^{2}}{g}\right\rfloor .\rho \left[\frac{g}{{cm}^{3}}\right]}$$

The apparent particle diameter obtained (Equation (4)) provides a similar value of 35 nm for E341(iii) A and C. E341(iii) B has an apparent diameter of 220 nm. The obtained values are only indicative and should be confirmed by the particle diameter obtained through microscopy imaging methods. Indeed, the particle shape may differ greatly from the spherical shape that the apparent diameter supposes.

### 3.3. Nanoparticle Size and Shape from Transmission Electron Microscopy

Transmission electron microscopy enables the determination of particle size and shape (Figure 7). HA in a powder form (bare particles) has a high tendency to form agglomerates once placed in a solvent (e.g., water) [25]. This makes the proper size distribution difficult to ascertain because of the prevalence of overlapping particles in the images. Unsuccessful attempts were made to break the agglomerates into single particles by using physical methods (ultrasonic bath—a less energetic method with unlikely particle alteration—and an ultrasonic probe—a more energetic method with possible particle alteration) and chemical methods (changing the solvent and introducing surfactants) (Appendix A). The combination of both physical and chemical separation methods did not yield proper particle separation. This implies that the particle–particle interaction is quite strong and that the system behaves as agglomerates or aggregates of particles.

E341(iii) A presents two types of structures: elongated particles of HA and micrometric plates corresponding to DCPA (Figure 8). E341(iii) B has much larger particles of irregular spherical shape. E341(iii) C, which has nanoparticle sizes closest to E341(iii) A, presents elongated shapes. 

The sizes of the nanoparticles were estimated on a series of TEM images similar to Figure 7 with agglomerated particles. The count procedure was carried out for 300 particles per sample by considering visible particles among the agglomerates in accordance to a counting procedure (counting rule 4) described by Bresch et al. [26]. This procedure consists in the consideration of agglomerated particles, which may overlap other particles. Counting is performed manually across multiple images, with the main counting criteria being particle visibility. The maximal and minimal Feret diameter (Dmax and Dmin, respectively) were measured. The aspect ratio (Dmax/Dmin) was determined for each individual particle and resulted in an aspect ratio distribution allowing a better appreciation of the general particle shape. The different size distributions have been fitted with a Gaussian model to help the visualization of the data (Figure 9).

The general statistical data of the distributions are reported in Table 3. E341(iii) A and C are most similar in terms of Dmin and Dmax, with a slightly broader distribution for E341(iii) C. The micrometric DCPA plates present in E341(iii) A were disregarded for the count (a proportion of 18% from the XRD analysis). E341(iii) B presents much larger Dmin and Dmax than both other samples. To evaluate the nanometric nature of the three samples, the proportion of particles below the 100 nm threshold was provided. On the one hand, the proportion of Dmax lower than 100 nm of the three samples ranged from 61% for E341(iii) B upwards to 90% for E341(iii) C and 94% for E341(iii) A. On the other hand, the Dmin of the particles, which is the smallest measured dimension for anisotropic particles, has been retained by EFSA as the most significant parameter to classify objects as nanomaterials: if one dimension is under 100 nm, then the sample is defined as a nanomaterial. E341(iii) A and C present all their HA particles with a Dmin smaller than 100 nm. Let us note that, in the case of E341(iii) A, its content in DCPA (micrometric plates) makes it 82% nanometric overall. For E341(iii) B, the proportion of particles with a Dmin lower than 100 nm is 89%.

The shape of the particles was estimated using their aspect ratio. The extremity of an oblong particle may influence the particle shape categorization (e.g., a pointy extremity for needle particles). On average, E341(iii) A has the largest aspect ratio of 3.3. Hence, E341(iii) A particles tend to be needle-like-shaped particles. E341(iii) B has an average aspect ratio of 2.0, which is consistent with the pseudo-spherical shape. E341(iii) C, with its average aspect ratio of 2.6, will be considered as rod-shaped particles. While some authors have provided guidelines, the association of a particle shape with a given aspect ratio can often rely on general appreciation [27].

### 3.4. Solubility as a Function of pH

Solubility measurement as a function of pH (Figure 10) is a key parameter of whether or not E341(iii) can be problematic when ingested, since gastric acidity can dissolve nanomaterials. In basic conditions, E341(iii) dissolution is negligible (at least 95% of remaining solid). From a pH of 6 to lower pH values, dissolution becomes more and more significant, and dissolution is complete below a pH of 2. The overall dissolution curves are very close for the three samples. From a pH of 6 to lower pH values, dissolution becomes more significant, and dissolution is complete below a pH of 2. Multiple parameters, including nanoparticle size and shape, composition, and surface area, can influence nanoparticle dissolution [28]. The dissolution of E341(iii) A was seemingly not influenced by its DCPA secondary phase, as DCPA presents a dissolution rate similar to HA (Appendix A).

Additionally, the state of the remaining solid in acidic conditions was analyzed by TEM. The images of the remaining solids were taken after undergoing acidic conditions (pH of 3, 48 h) (Figure 11).

In all samples, nanoparticles were still present. The E341(iii) A and C particles are very similar in appearance and with significantly fewer needle-shaped particles than in the pristine samples. The E341(iii) B sample displays different populations of nanoparticle sizes (smaller than in its pristine form), as if different stages of dissolution were simultaneously present. The overall shape is still spherical-like, but seemingly here closer to a spherical shape. Additionally, the E341(iii) B sample also shows particle coalescence as well as the presence of a large number of very small nanoparticles (Figure 12).

### 3.5. Agglomerates Size

Laser granulometry performed on the E341(iii) samples in neutral conditions (Figure 13) shows that E341(iii) tends to form agglomerates in aqueous conditions. Agglomerate sizes are mostly around 10 µm for all the samples. Larger sizes may be observed for E341(iii) A and C, which seem to have a significant proportion of agglomerates, respectively, in the 100 µm and 50 µm range.

### 3.6. Surface Charge in Aqueous Solution

Zeta-potential measurements were performed on the three E341(iii) samples, according to pH, in order to determine the evolution of the surface charge all along the digestive media. At this stage, ionic strength was fixed by NaCl salt, but no additional species were introduced in the medium, even if the surface charge can be greatly affected by the presence of salts or proteins. 

Due to the dissolution that occurs in acidic conditions, the zeta potential measured below a pH value of 6 is not necessarily representative of the starting material. However, the presence of residual solid particles in these conditions allowed zeta-potential measurements with highly repeatable values (Figure 14).

The three E341(iii) samples displayed similar zeta-potential behavior as a function of the pH. Namely, the negative values of the zeta potential decreases from a pH of 6 to higher pH values (trend 1). From a pH of 6 to a pH of 4 (trend 2), a decrease in the zeta potential is observed before a rise starting from a pH of 4 to lower pH values (trend 3). Below a pH a 6, the particles are submitted to dissolution, which becomes more and more significant as more acidic conditions are reached. The phosphate and calcium ions constitutive of HA start to be released in the solution and increase in concentration as the pH decreases (Figure 10). Therefore, the aqueous medium is altered in acidic conditions, which affects the zeta potential due to surface interactions with the released ions.

## 4. Discussion

Even if the three E341(iii) samples exhibit notable differences (Table 4), interestingly, all the E341(iii) samples were essentially composed of HA. Hence, although the indication of TCP for E341(iii) food additive suggests it would chemically be Ca_3_(PO_4_)_2_, its chemical composition actually corresponds to hydroxyapatite, with the true formulation Ca_5_(PO_4_)_3_(OH). This conclusion has been reached from the XRD diffraction patterns, and was also confirmed by solid state ^31^P NMR, FT-IR, and Raman spectroscopy. The E341(iii) A sample is the only one that presented a secondary phase of DCPA, which consisted in micrometric plates with a mass proportion of 18%. The size and shape of the HA particles were different for the three samples, even if E341(iii) A and C present the closest morphology characteristics. E341(iii) A particles were identified as needle-like due to their larger aspect ratio (3.3) compared to that of E341(iii) C particles (2.6), which were categorized as rod-shaped. These ratios are slightly smaller than those observed for HA nanoparticles characterized as needle-like in the previous work published on the subject, where an aspect ratio was determined at a minimum value of 4.2 [8]. Ascribing a particle shape name to a particle does not necessarily rely only on the particle aspect ratio, but also on other parameters, such as the general particle aspect and oblong particle extremity shape, which may influence the labeling [29]. E341(iii) B particles, with an aspect ratio of 2.0, present shapes described as pseudo-spherical. Additionally, E341(iii) B presents much larger dimensions than E341(iii) A and C.

Concerning the nitrogen adsorption measurements, the C_BET_ constant was different for all three samples (E341(iii) A > E341(iii) C > E341(iii) B). This could be representative of the differences in particle shape, and thus in the expressed crystal facets. The VSSA was comparable for E341(iii) A and C (166 m^2^/cm^3^ and 178 m^2^/cm^3^, respectively), but notably lower for E341(iii) B (27 m^2^/cm^3^). Indeed, since a larger particle size implies a lower specific surface area, the larger particle size is the main parameter explaining this last lower E341(iii) B value. As expected, this trend is confirmed with E341(iii) A and C (58 m^2^/g and 60 m^2^/g, respectively), having a specific surface area more than six times as great as E341(iii) B (9 m^2^/g). 

The identity of E341(iii), also referred to as simply E341, tricalcium phosphate or salts of phosphoric acid, could be established. The three considered samples exhibited enough differences so that some of the possible variations of E341(iii) could be explored. The large difference of C_BET_ values reveals that the surface chemical sites are different and will have to be investigated by surface physical techniques in the future. However, there were also sufficient similarities in order to define the general characteristics of this food additive: E341(iii) is nano-HA (Table 5). The most notable differences between samples were the large particle size of E341(iii) B and the presence of a secondary phase of DCPA for E341(iii) A.

The VSSA criterion is also a good indicator of the nanometric nature of E341(iii) [30], as samples A and C exceed the 60 m^2^/cm^3^ threshold. Despite not fulfilling the VSSA criterion, E341(iii) B cannot be ruled out as being nanometric on the basis of this single criterion. Indeed, while the population of particles with a Dmax exceeding 100 nm is 39%, the proportion of particles with a Dmin below 100 nm is 89%. Moreover, according to European Union regulations, a substance containing 50% or more particles with a dimension in the 1 nm–100 nm size range is considered as nanoform, regardless of the agglomerated or aggregated state of the particles [1]. Therefore, all three E341(iii) samples analyzed in our work are nanomaterials (Table 3). 

The HA nanoparticles constitutive of the E341(iii) samples form agglomerates of mostly about 10 µm in aqueous solution (Figure 13). This means that, although the smallest primary particles are nanometric in size, the particles are present in the form of large agglomerates. As a consequence, the agglomerated HA nanoparticles tend to have a fast sedimentation rate with no suspended particles within a few hours (Appendix A).

Nanomaterials can be dismissed as being of no safety concern if they are completely dissolved in acidic conditions similar to gastric conditions [13]. Hence, the dissolution of E341(iii) with respect to pH was studied (Figure 10). Here, it was found that E341(iii) dissolves under a pH of 6 in a gradual manner, with particles still present at a pH of 3 (Figure 11 and Figure 12). Dissolution starts from a pH of 6 to lower pH values and happens in a gradual manner, with particles still present at a pH of 3 (Figure 10 and Figure 11).

The zeta-potential measurements (Figure 14) showed that the HA nanoparticles constitutive of E341(iii) were mostly negatively charged on the pH range, where dissolution was negligible (pH > 6). The surface became more and more negatively charged with the increasing pH (trend 1, Figure 14). This is explained by the phosphate surface groups, which are deprotonated in basic conditions. The interpretation of the trends observed below a pH of 6 (trends 2 and 3, Figure 14) are more difficult to assess, as there is no control over the ions present in the aqueous medium. Indeed, due to partial dissolution, HA particles release phosphate and calcium ions with an increasing concentration as the pH value goes down. Regarding the increase in the zeta potential observed from a pH of 4 to a pH of 2 (trend 3), it is likely due to the adsorption of calcium ions onto the particle surface.

From the overall collected information on the E341(iii) food additive, a clearer picture of its physicochemical nature can be drawn, along with elements of its behavior in aqueous conditions. Notably, the actual proportion of nanometric E341(iii) remaining after a digestion cycle cannot be predicted solely from this study, as the experimental conditions used here are quite simple compared to a digestion process. The reactivity of HA nanoparticles, which constitute E341(iii), are expected to be more complex in a digestive setting. The fate of the HA nanoparticles during digestion is an important issue, as they are not subject to complete dissolution in gastric conditions. Furthermore, the adsorption of enzymes and salts onto HA nanoparticles could prevent dissolution entirely by forming a protective material layer [31]. Moreover, the undissolved HA nanoparticles should come into interaction with the gastrointestinal barrier, questioning their potential permeation through this barrier, as is the case for other materials of nanometric dimensions [32]. Understanding how the E341(iii) food additive is addressed during digestion will be the next step in establishing its potential toxicity. 

## 5. Conclusions

In this study, we present the first detailed characterization of the E341(iii) food additive. Also called tricalcium phosphate, the E341(iii) food additive is actually composed of hydroxyapatite (HA), and the nanometric nature of all samples has been shown. This resulted in the conclusion that the E341(iii) food additive, represented by three independent samples, satisfies the definition of a nanomaterial according to the EFSA. The conclusions given in a previous work, conducted in the United States and evidencing HA nanoparticles in infant formula, are thus consistent with the materials constitutive of E341(iii) used as food additive in Europe [8].

In addition, we have shown that E341(iii) displays various particle forms ranging in aspect ratio from needle-like to pseudo-spherical shapes. In a simple aqueous medium, the HA nanoparticles formed agglomerates of mostly 10 µm, leading to unstable suspensions—complete sedimentation occurs within hours. These agglomerates disappear progressively once dissolution starts below a pH of 6. The characterization of the E341(iii) food additive conducted in this work will provide relevant information to consider its reactivity in the more complex digestion process. Complete dissolution below a pH of 2 hints that, in the gastrointestinal tract, nanoparticles of HA may still be present, raising the question of their fate during digestion.

## Figures and Tables

**Figure 1 nanomaterials-13-01823-f001:**
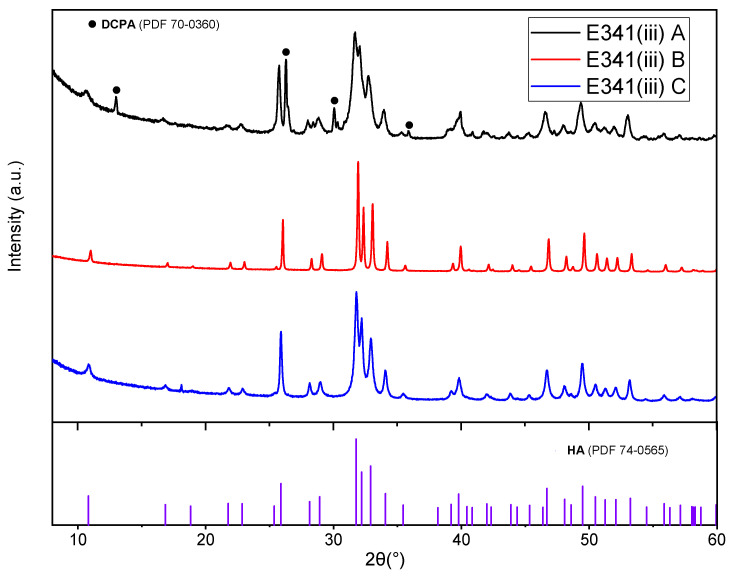
X-ray diffraction patterns of pristine food-grade E341(iii). The peak assignment of E341(iii) A, B, and C was made using references for HA (PDF 74-0565) and DCPA (PDF 70-0360).

**Figure 2 nanomaterials-13-01823-f002:**
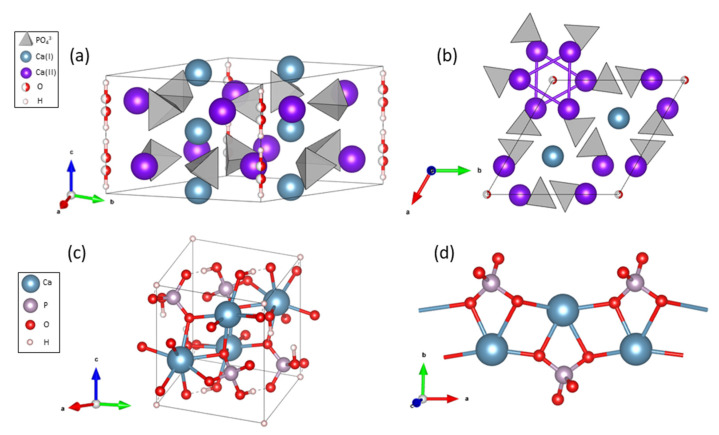
Structure of HA with (**a**) its hexagonal lattice and (**b**) a view from the c-axis showing the channel organization encompassing hydroxyl ions. Structure of DCPA with (**c**) its triclinic lattice and (**d**) the Ca-PO_4_ double chain organization. Structures were drawn from cif file numbers 9,011,093 (HA) and 9,007,619 (DCPA).

**Figure 3 nanomaterials-13-01823-f003:**
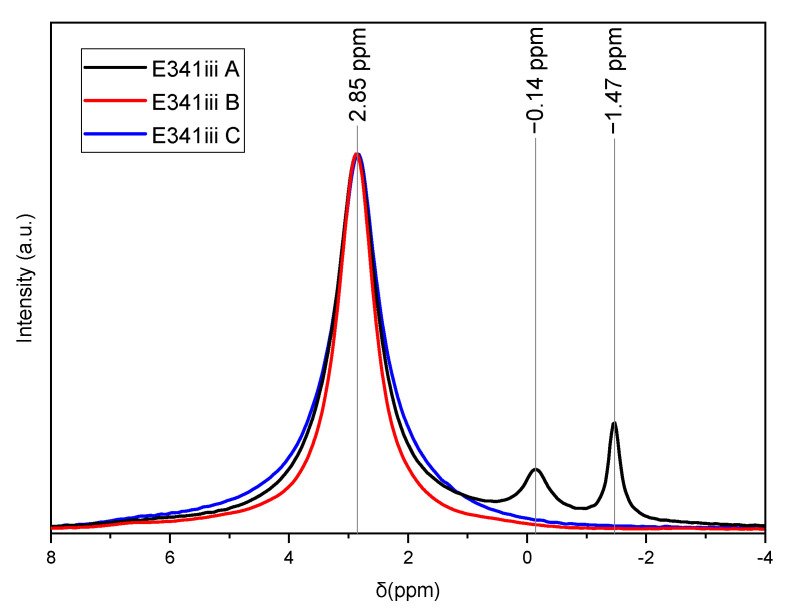
Solid State ^31^P NMR spectra of E341(iii) A, B, and C showing that all E341(iii) samples are constituted of HA (2.85 ppm). E341(iii) A contains a secondary DCPA phase (−0.14 and −1.47 ppm).

**Figure 4 nanomaterials-13-01823-f004:**
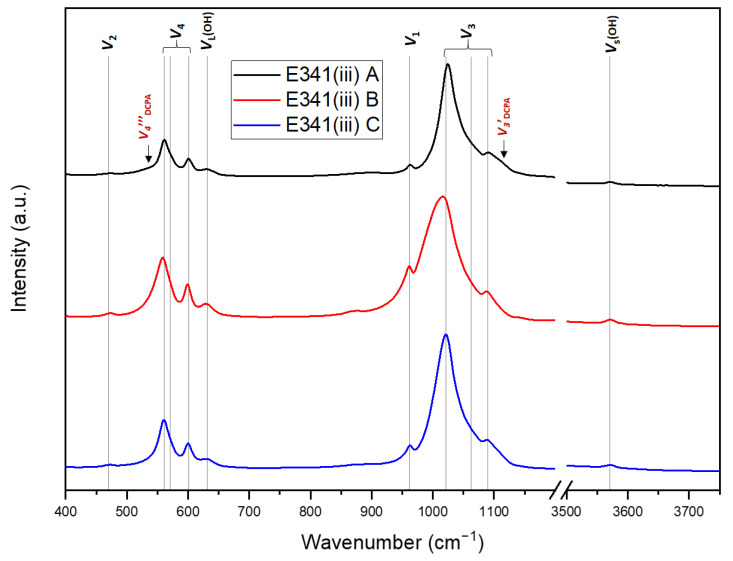
FT-IR spectra (ATR) of E341(iii) A, B, and C with the assigned hydroxyapatite vibration modes and for E341(iii) A, two DCPA vibration modes.

**Figure 5 nanomaterials-13-01823-f005:**
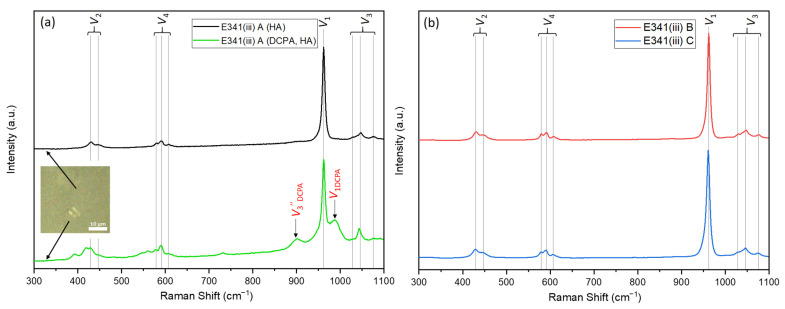
Raman spectra (514 nm) of (**a**) E341(iii) A with the spectra of both crystallographic phases (optical microscopy image in insert) present in the sample; (**b**) E341(iii) B and C.

**Figure 6 nanomaterials-13-01823-f006:**
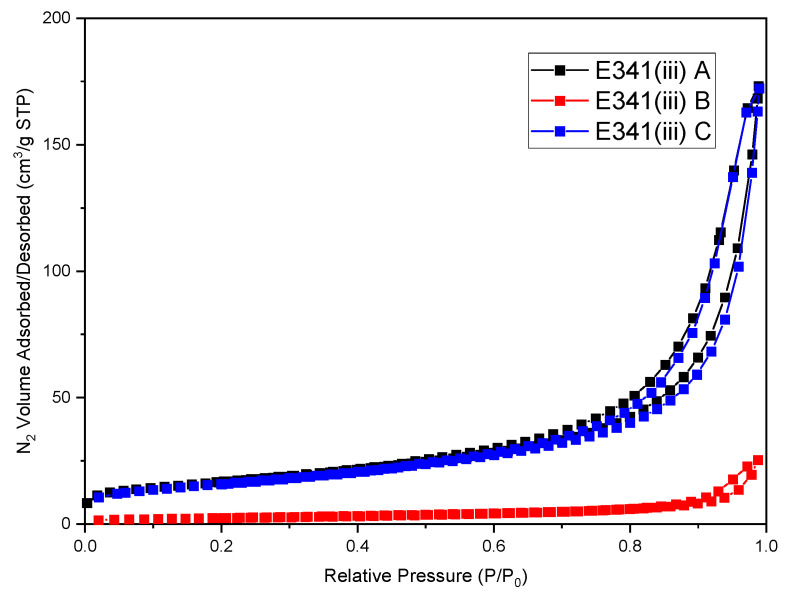
N_2_ adsorption/desorption isotherm of the E341(iii) samples.

**Figure 7 nanomaterials-13-01823-f007:**
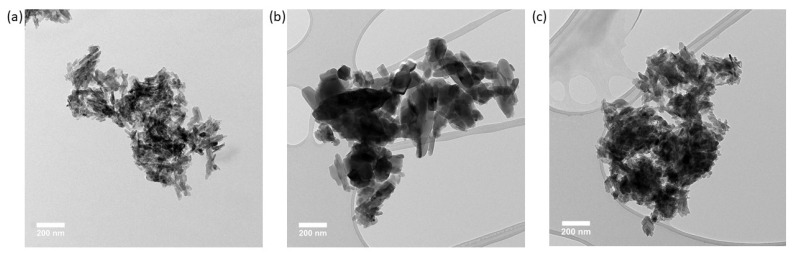
TEM Images of (**a**) E341(iii) A, (**b**) E341(iii) B, and (**c**) E341(iii) C.

**Figure 8 nanomaterials-13-01823-f008:**
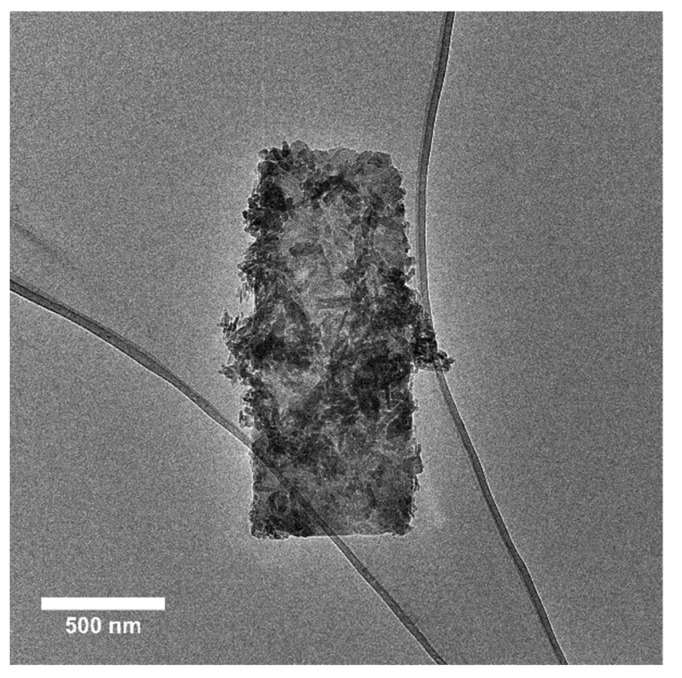
Micrometric plate of DCPA present in sample E341(iii) A, here covered with HA nanoparticles.

**Figure 9 nanomaterials-13-01823-f009:**
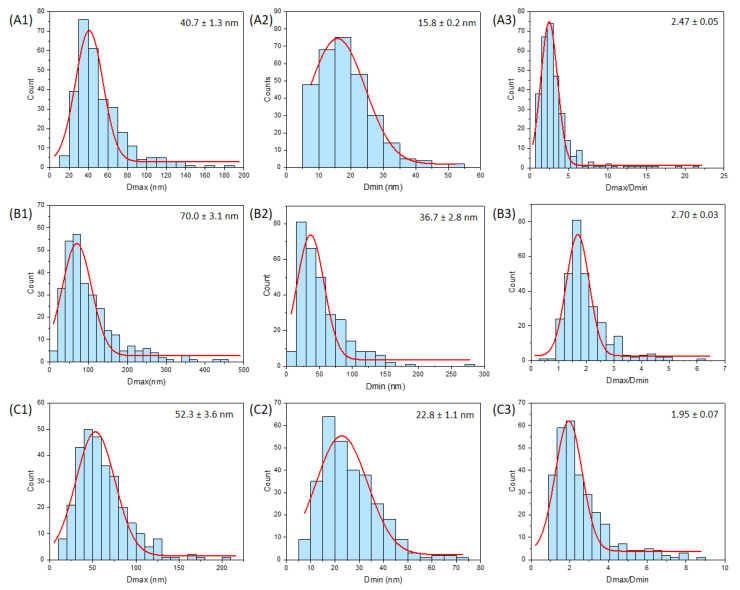
Particle size distribution of E341(iii) A for its Dmax (**A1**) and Dmin (**A2**) and its particle aspect ratio distribution (**A3**); E341(iii) B for its Dmax (**B1**) and Dmin (**B2**) and its particle aspect ratio distribution (**B3**); E341(iii) C for its Dmax (**C1**) and Dmin (**C2**) and its particle aspect ratio distribution (**C3**). The red curve corresponds to the Gaussian fit applied (average result presented in the upper right-hand corner of each plot).

**Figure 10 nanomaterials-13-01823-f010:**
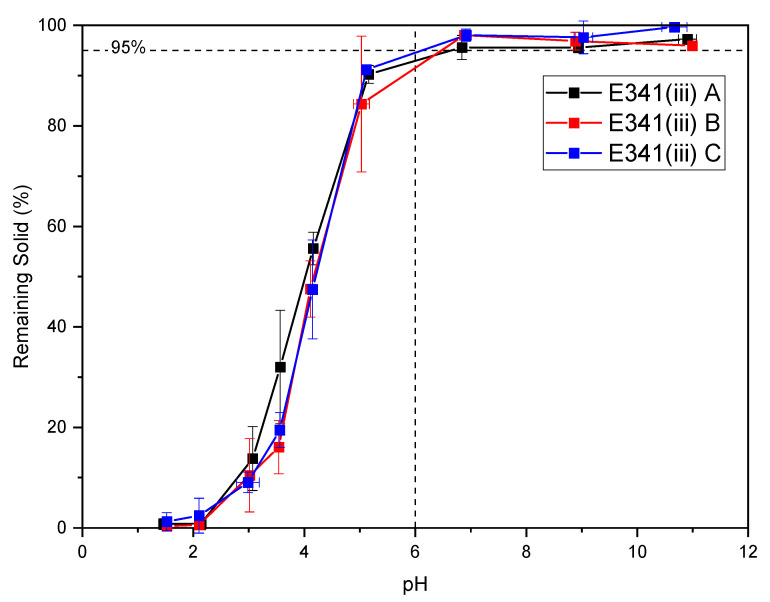
Remaining solid after placing 100 mg (initial mass) of E341(iii) in 20 mL of ultrapure water at different pH values for 48 h at room temperature with moderate stirring (40 rpm). Dash line indicates when remaining solid starts to dissolve (under 95% remaining solid), corresponding to a pH of 6.

**Figure 11 nanomaterials-13-01823-f011:**
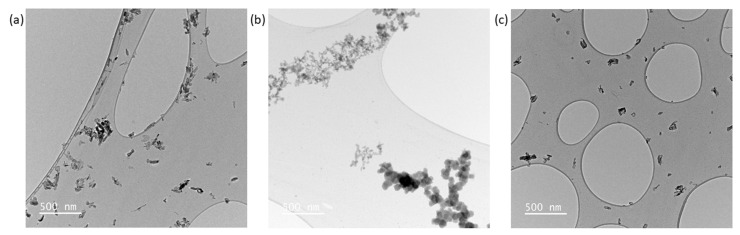
TEM images of (**a**) E341(iii) A, (**b**) B, and (**c**) C samples after 48 h at a pH of 3.

**Figure 12 nanomaterials-13-01823-f012:**
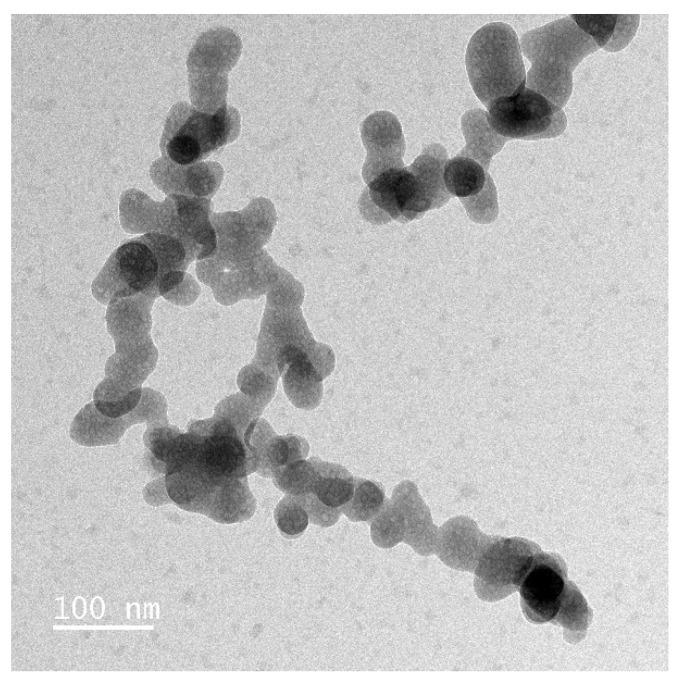
TEM image of E341(iii) B (pH of 3, 48 h) with coalesced particles and very small nanoparticles present throughout the sample.

**Figure 13 nanomaterials-13-01823-f013:**
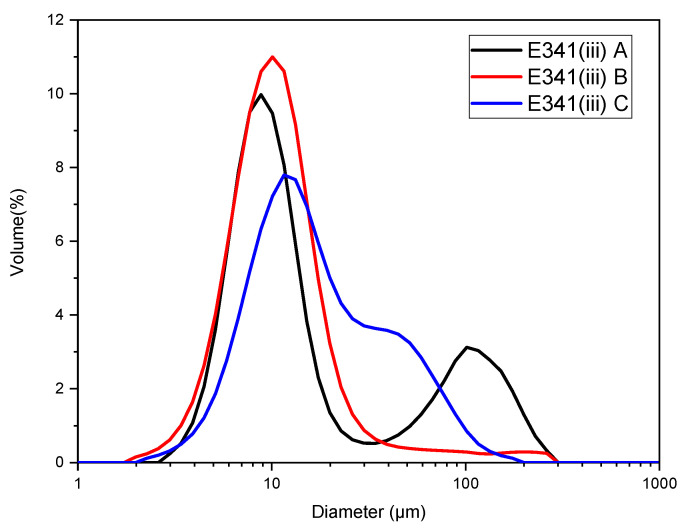
Diameter of agglomerates of E341(iii) A, B, and C in neutral pH conditions (for a pH of 6.87, 7.00, and 6.47, respectively) at room temperature.

**Figure 14 nanomaterials-13-01823-f014:**
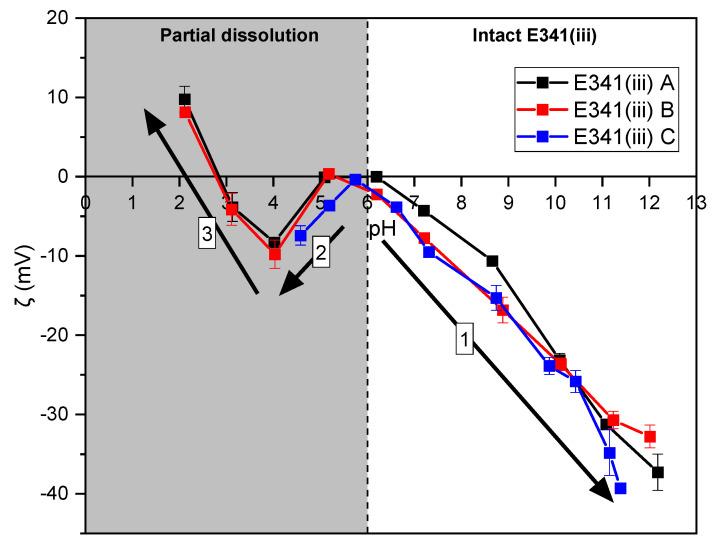
Zeta-potential measurements of E341(iii) A, B, and C according to pH. Although zeta-potential values are reliable below a pH of 6, the dissolution phenomenon of E341(iii) starts to occur (gray area). Three trends (numbered arrows) are observed.

**Table 1 nanomaterials-13-01823-t001:** Atomic Ca/P molar ratio (center column) and molar mass ratio (right column) of pristine E341(iii) samples determined by ICP-AES.

Sample	Ca/P (Atomic)	Ca/P (Mass)
E341(iii) A	1.52 ± 0.02	1.97 ± 0.03
E341(iii) B	1.71 ± 0.02	2.21 ± 0.03
E341(iii) C	1.71 ± 0.03	2.21 ± 0.04

**Table 2 nanomaterials-13-01823-t002:** Specific surface area and C_BET_ constants obtained from the BET analysis of the adsorption desorption isotherms performed with N_2_ gas, density obtained through He pycnometry, and calculated VSSA and apparent particle diameter, d_BET,App_.

Sample	Specific Surface Area Sw (m^2^/g)	C_BET_ (N_2_)	ρ (g/cm^3^)	VSSA (m^2^/cm^3^)	d_BET,App_ (nm)
E341(iii) A	58	229	2.9	166	35
E341(iii) B	9	113	3.0	27	220
E341(iii) C	60	150	2.9	178	35

**Table 3 nanomaterials-13-01823-t003:** Statistical parameters obtained from the TEM size distribution of the three E341(iii) samples. Q1 and Q3 are, respectively, the first quartile and third quartile. N represents the percentage of particles with a dimension (Dmin and Dmax, respectively) below the 100 nm threshold. Average parameters are indicated with their standard deviation.

	Parameter	E341(iii) A	E341(iii) B	E341(iii) C
Dmax	Average (nm)	52 ± 26	102 ± 73	62 ± 29
	Median (nm)	44	81	56
	Q1 (nm)–Q3 (nm)	33–62	52–128	40–77
	N (D < 100 nm) (%)	94	61	90
Dmin	Average (nm)	18 ± 8	54 ± 35	26 ± 12
	Median (nm)	18	43	23
	Q1 (nm)–Q3 (nm)	12–23	28–69	18–33
	N (D < 100 nm) (%)	100	89	100
Dmax/Dmin	Average	3.3 ± 2.7	2.0 ± 0.8	2.6 ± 1.4
	Median	2.8	1.8	2.2
	Q1–Q3	1.9–3.7	1.5–2.3	1.6–3.1
	Associated shape	Needle-like	Pseudo-spherical	Rod

**Table 4 nanomaterials-13-01823-t004:** Summary of the main characteristics of the three E341(iii) samples, evidencing the differences between them.

Parameter	E341(iii) A	E341(iii) B	E341(iii) C
Phase	HA and DCPA	HA	HA
Average Dmax (nm); Dmin (nm)	52 ± 26; 18 ± 8	102 ± 73; 54 ± 35	62 ± 29; 26 ± 12
HA particle shape (Average Dmax/Dmin)	Needle-like (3.3 ± 2.7)	Pseudo-spherical (2.0 ± 0.8)	Rod (2.6 ± 1.4)
C_BET_ (N_2_)	229	113	150
VSSA (m^2^/cm^3^)	166	27	178
Sw (m^2^/g)	58	9	60

**Table 5 nanomaterials-13-01823-t005:** Identity card of E341(iii), based on the analyzed samples, and associated techniques used in this study.

Properties	Technique	E341(iii) Identity
Chemical composition/identity	- Inductively coupled plasma–optical/atomic spectroscopy (ICP-OES/AES)	Ca and P elements with a Ca/P ratio consistent with hydroxyapatite
Shape	- Transmission electron microscopy (TEM)	Different possible shapes (needle, rod, spherical)
Particle size and size distribution; Agglomeration/Aggregation state	- TEM - Laser granulometry	Primary nanometric particles in mostly 10 µm agglomerates
Crystal form and phase	- XRD - Infrared and Raman spectroscopy - Nuclear magnetic resonance spectroscopy (NMR)	E341(iii) is mainly HA
Surface area (volume, mass specific)	- Adsorption isotherms methods with Brunauer–Emmett–Teller method (BET)	Moderate (58–60 m^2^/g) to possibly low values (9 m^2^/g)
Surface chemistry	- BET model for nitrogen gas adsorption	C_BET_ constant dependent on surface affinity for N_2_ (function of size and particle shape)
Surface charge	- Electrophoretic light scattering (ELS)/zeta potential	Negatively charged above a pH of 6
Degradation/ Dissolution/Solubility	- Dissolution as a function of the pH	Dissolution under a pH of 6

## Data Availability

Not applicable.

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
