# Peer review of "The True Nature of Tricalcium Phosphate Used as Food Additive (E341(iii))"

_nanomaterials, 2023, doi:10.3390/nano13121823_

Round 1
Reviewer 1 Report
The author´s aim was to determine whether Tricalcium Phosphate food additive (E341(iii)) as is used in Europe can be classified as a nanomaterial and to identify its properties suitable for understanding its oral consumption implications. They concentrate on characterizing both its crystallographic structure and its physicochemical properties. Agencies such as the European Food and Safety Authority (EFSA) provide guidelines to identify materials exhibiting nanoscale dimensions and their physicochemical properties. These recommendations are taken into account to conduct this study. The results of this study point out that E341(iii) food additive, represented by three independent samples, is in fact a nanomaterial. The characterization of E341(iii) food additive in this work provides relevant information to consider its reactivity in the more complex digestion process. The manuscript reviews 32 articles regarding this topic. The topic of this manuscript is up‑to‑date, attractive and well‑suited for your journal. The manuscript is well-written and divided into 5 main parts, the text is clear and easy to read. For better visualisation, the authors used 4 tables and 14 figures. I suggest checking for some spelling mistakes and grammar errors. Otherwise, I have no major concerns about this manuscript and I recommend it for publication.
I suggest checking for some spelling mistakes and grammar errors.
Author Response
The reviewer feedback was carefully considered. All authors have seen and approved the submission of the reviewed manuscript.
Please see the attachment.

Reviewer 2 Report
Review Nanomaterials-2407627
The work is useful and the authors provided comprehensive parameters/results to support the statement/hypothesis. However, some places were not clear enough to elaborated the ideas which should be improved.
The FTIR results were not visually good enough to tell the difference among the groups. It is suggested to highlight the fingerprinting zone/wavenumber areas and enlarge that part. Thus, the difference among the groups will be visually clear. For instance, Food Hydrocolloids, 75, 164-173.
The authors highlighted gastro-intestinal effect might be caused by this compound. However, in the introduction and discussion, the corresponding description was missing. The gastro-intestinal modulation on compound should be mentioned, say by a literature review to fill the gap between the results and the possible gastro-intestinal effects. For instance, Food Chemistry, 399, 133959.
NMR result: what was the internal standard applied, which should be labeled in the Figure and described in the section methodology. For instance, Food Chemistry, 286, 87-97.
The integration of the results from different parameters can be enhanced. Some of the results can be put as supplemental data thus the Figures/Tables in the full text can be more integrated.
Author Response

(The authors gave the same response as above.)

Reviewer 3 Report
The paper by Humbert, B. et al. entitled "The True Nature of Tricalcium Phosphate used as Food Additive (E341(iii))" considers a very practical aspect of food additives involving an impressive number of very sophisticated analytical techniques such as ICP-OES/AES, TEM, XRD - Infrared and Raman spectroscopy - NMR, (ELS)/zeta potential and others. The topic concerns food additives and is therefore practical and necessary. The paper is very well written and perfectly illustrated. I recommend this paper for publication. Before publication, I suggest that the following issues be considered:
1. Please change the chemical names to the following: Calcium phosphate monobasic, Calcium phosphate dibasic, Calcium phosphate tribasic
2. As A341 sigma aldrich actually offers at least 10 products of hydroxyapatite - so please give more precise information about the studied product A as well as B and C.
3. Table 1 Adding the mass ratio of Ca/P
4. The 268 HA/DCPA ratio obtained by this method (78:22) is similar to that previously obtained by the Rietveld method (82:18)" - To which sample does the above 78:22 ratio apply?
5. Lines 320-322 The authors wrote: "E341(iii) A shows two types of structures: elongated particles of HA and micrometric plates corresponding to DCPA (Figure 8). E341(iii) B has much larger particles of irregular spherical shape. E341(iii) C, which has nanoparticle sizes closest to E341(iii) A, has elongated shapes" - It is difficult to see two types of structure in sample A. How did you determine that HA particles are elongated and DCPA particles are micrometric? Could you please explain. The same question applies to Fig.8
6. Fig. 11/12- You can reduce the number of figures showing the same magnification (100 nm).
7. Lines 431-435 are repetitions of lines 418-422.
8. Please explain trend 1 in Fig.14
9. Why did you not use ELS for size distribution?
10. The conclusion should be shortened, instead some details should be added in the abstract.
Author Response

(The authors gave the same response as above.)
